# An intranasal live-attenuated SARS-CoV-2 vaccine limits virus transmission

Julia M. Adler [1], Ricardo Martin Vidal [1], Christine Langner [1], Daria Vladimirova[1], Azza Abdelgawad[1], Daniela Kunecova[1], Xiaoyuan Lin [1,2], Geraldine Nouailles [3], Anne Voss[4], Sandra Kunder [4], Achim D. Gruber [4], Haibo Wu [2], Nikolaus Osterrieder[1], Dusan Kunec[1,5] & Jakob Trimpert [1,5] ✉

The development of effective SARS-CoV-2 vaccines has been essential to control COVID-19, but significant challenges remain. One problem is intramuscular administration, which does not induce robust mucosal immune responses in the upper airways—the primary site of infection and virus shedding. Here we compare the efficacy of a mucosal, replication-competent yet fully attenuated virus vaccine, sCPD9-ΔFCS, and the monovalent mRNA vaccine BNT162b2 in preventing transmission of SARS-CoV-2 variants B.1 and Omicron BA.5 in two scenarios. Firstly, we assessed the protective efficacy of the vaccines by exposing vaccinated male Syrian hamsters to infected counterparts. Secondly, we evaluated transmission of the challenge virus from vaccinated and subsequently challenged male hamsters to naïve contacts. Our findings demonstrate that the live-attenuated vaccine (LAV) sCPD9-ΔFCS significantly outperformed the mRNA vaccine in preventing virus transmission in both scenarios. Our results provide evidence for the advantages of locally administered LAVs over intramuscularly administered mRNA vaccines in preventing infection and reducing virus transmission.

The COVID-19 pandemic has had a profound impact on human health, resulting in millions of deaths and causing widespread disruption of daily life globally. The ongoing circulation of SARS-CoV-2 enables the constant evolution of the pathogen, presenting continuous challenges to both individual and public health. One of the most significant achievements in the fight against this pandemic has been the rapid development of effective vaccines against SARS-CoV-2. While the developed vaccines have been successful in reducing the severity of illness of the upper and lower respiratory tract and, in particular, the number of deaths associated with virus infection, significant challenges remain.

One of the primary challenges is the inadequacy of the current vaccines in providing effective protection against SARS-CoV-2 infection[1–5]. Additionally, the virus's ability to mutate and evolve

leading to more effective transmission and immune evasion, further jeopardizes the efficacy of current vaccines[2–4]. This challenge is particularly evident with the emergence of the highly transmissible Omicron variant and its numerous subvariants, which are only marginally controlled by available vaccines[2,3]. This highlights the importance of developing vaccines that can effectively prevent infection, reduce transmission, and potentially limit virus evolution.

The focus of our research has been the development of a live-attenuated vaccine (LAV) candidate sCPD9[6–10]. This vaccine candidate was developed by codon pair deoptimization (CPD)[8]. CPD rearranges synonymous codons within the viral genome, which results in attenuation but completely maintains viral protein sequences[11]. In the case of sCPD9, recoding of a distinct sequence at the 3′ end of SARS-CoV-2 ORF1ab[6] yielded a replication-competent

[1]Institut für Virologie, Freie Universität Berlin, Berlin, Germany. [2]School of Life Sciences, Chongqing University, Chongqing, China. [3]Department of Infectious Diseases, Respiratory Medicine and Critical Care, Charité—Universitätsmedizin Berlin, corporate member of Freie Universität Berlin and Humboldt-Universität zu Berlin, Berlin, Germany. [4]Institut für Tierpathologie, Freie Universität Berlin, Berlin, Germany. [5]These authors jointly supervised this work: Dusan Kunec, Jakob Trimpert. ✉e-mail: jakob.trimpert@fu-berlin.de

and fully attenuated vaccine virus, in which the full antigenic repertoire of SARS-CoV-2 is preserved and is applied intranasally[8]. Owing to these properties, sCPD9 induces strong mucosal immunity in the respiratory tract as well as systemic immunity against a range of viral antigens, which sets it apart from intramuscularly administered, spike-based mRNA and vectored vaccines[10]. However, LAVs can pose a risk of unintended virus spread. To address this concern, we have generated a non-transmissible sCPD9 version, called sCPD9-ΔFCS, by removing the furin cleavage site from the viral spike protein[9]. Remarkably, sCPD9-ΔFCS, when administered to Syrian hamsters (*Mesocricetus auratus*), is completely non-transmissible, yet provides the same level of protection against SARS-CoV-2 as the original sCPD9 virus[9].

Intranasal vaccines offer several advantages over intramuscular vaccines, mainly by eliciting an immunoglobulin A (IgA) response and establishing resident memory B and T cells in the respiratory mucosa[12]. These two layers of protection provide an effective barrier to infection, restricting virus replication, virus shedding, and transmission from the respiratory tract more effectively than intramuscular vaccines that induce only systemic immunity.

In this study, we compared the efficacy of the LAV candidate sCPD9-ΔFCS and the monovalent mRNA vaccine BNT162b2 in preventing the spread of two wild-types (WT) viruses: the variant B.1 and the Omicron subvariant BA.5. We examined their performance in two distinct scenarios: (1) protecting vaccinated individuals from contracting the disease after exposure to WT-infected shedders; and (2) preventing the transmission of the virus to naive contacts when vaccinated individuals were infected with WT virus 35 days after vaccination. Importantly, both vaccines are based on the SARS-CoV-2 B.1 variant, allowing for a head-to-head comparison of their efficacy against homologous and heterologous virus challenges. Our findings demonstrate that the attenuated virus vaccine sCPD9-ΔFCS significantly outperformed the mRNA vaccine in preventing virus transmission in both scenarios. In addition, we show clear benefits of intranasally administered vaccines against replicating viruses in terms of preventing infection, reducing virus transmission, and limiting further virus evolution.

## Results

### LAV and mRNA vaccine prevent the development of clinical symptoms following natural transmission of SARS-CoV-2 B.1 and BA.5

In the first set of experiments, Syrian hamsters received two doses of sCPD9-ΔFCS, BNT162b2, or a mock vaccine at an interval of 3 weeks. After fourteen days, the hamsters were brought in contact with animals that had been infected with either the virus variant B.1, or the Omicron BA.5 one day prior. Two vaccinated animals were co-housed with one shedder hamster for 6 days and closely monitored for disease symptoms and virus loads in the upper airways (Fig. 1a).

All three groups of shedder animals infected with the SARS-CoV-2 variant B.1 showed moderate weight loss and clinical symptoms typical of COVID-19-like pneumonia (Fig. 1b and S1a). As expected, shedder animals infected with the BA.5 variant[13,14] showed less pronounced weight loss, with some variability among the infected groups (Fig. 1c and S1b).

The sCPD9-ΔFCS and mRNA vaccines effectively prevented body weight loss in vaccinated contact animals exposed to both B.1 or BA.5 shedders (Fig. 1c, b). As anticipated, mock-vaccinated animals exposed to B.1 shedders exhibited a progressive decline in body weight starting from day 3 after contact (Fig. 1b). Meanwhile, mock-vaccinated animals that were exposed to BA.5 shedders did not experience weight loss, likely due to the lower pathogenicity of the BA.5 variant and the comparatively milder disease outcome observed in infected animals (Fig. 1c).

### Only LAV prevents transmission of SARS-CoV-2 B.1 and BA.5

To monitor virus replication and transmission, oral swabs were collected daily during co-housing of infected and contact animals. Furthermore, oropharyngeal swabs and lungs were obtained on day 6 post contact (dpc) to quantify viral RNA levels and replicating virus (Fig. 1a). Infected shedder animals exhibited high levels of SARS-CoV-2 RNA with a gradual decrease towards day 6. In general, virus loads were higher in B.1 shedders compared to BA.5 shedders. Differences in SARS-CoV-2 copy numbers between individual shedders were slightly more pronounced, but still not significantly different in BA.5-infected animals (Figs. 1d, e and S1c–h; Table S1). No influence on onward transmission was observed.

Consistent with the observed body weight loss, oral swabs of mock-vaccinated contacts exposed to B.1 shedders had high levels of SARS-CoV-2 genomic RNA (gRNA) from 2 dpc onward (Fig. 1d). Although mRNA vaccination prevented body weight loss, it did not protect against B.1 infection, as evidenced by sustained high SARS-CoV-2 gRNA levels in oral swabs after day 2. Nevertheless, mRNA vaccination did result in lower viral gRNA levels compared to mock vaccination. In contrast, sCPD9-ΔFCS-vaccinated contacts displayed minimal SARS-CoV-2 gRNA levels, oscillating around the detection limit, suggesting repeated exposure to the virus without signs of productive infection in the upper airways (Fig. 1d).

Similar results were obtained in vaccinated hamsters exposed to BA.5-infected shedders. SARS-CoV-2 gRNA levels in oral swabs of mRNA- and mock-vaccinated animals peaked on dpc 4 to 5 (Fig. 1e), while virus gRNA levels in sCPD9-ΔFCS-vaccinated hamsters were near the detection limit, suggesting effective prevention of BA.5 infection (Fig. 1e).

Consistent with these findings, both mRNA- and mock-vaccinated hamsters exposed to B.1 or BA.5 shedders showed high SARS-CoV-2 gRNA levels in oropharyngeal swabs and lungs on 6 dpc (Fig. 1f, g). Additionally, the replication-competent virus was present in the lung tissue of mock-vaccinated animals exposed to B.1 shedders, while most shedders had cleared the infection by that time (Fig. 1f and S1g). Despite slightly lower viral gRNA levels, a replicating virus was found in the lungs of two mRNA-vaccinated and three mock-vaccinated hamsters exposed to BA.5 shedders, reflecting the delayed peaking of virus replication compared to B.1 infection (Fig. 1g).

Additionally, virus replication was monitored by quantifying the presence of subgenomic RNA (sgRNA) transcripts in daily oral swabs, as the presence of viral replicative RNA intermediates reliably indicates active virus replication[15–17]. Consistent with gRNA quantification results, sgRNA was detected in all mRNA- and mock-vaccinated contact animals that were co-housed with either B.1- or BA.5-infected shedders. In contrast, contact animals vaccinated with sCPD9-ΔFCS exhibited undetectable or very low sgRNA copy numbers, confirming the protective efficacy of sCPD9-ΔFCS vaccination against SARS-CoV-2 B.1 and BA.5 transmission (Fig. 2a, b and Table S2).

In contrast to the pneumonia observed in mock-vaccinated animals, pulmonary lesions were largely absent in sCPD9-ΔFCS-vaccinated and less pronounced in mRNA-vaccinated animals exposed to B.1 shedders. Hamsters experimentally infected with the B.1 virus or animals that contracted B.1 after mock vaccination developed lesions typical of COVID-19-like pneumonia (Fig. 3a and Fig. S2a, b). Specifically, the infected animals had pronounced patchy bronchointerstitial pneumonia with necrosuppurative bronchitis and bronchiolitis, the proliferation of alveolar type II epithelia, vascular endotheliitis, diffuse alveolar damage, as well as perivascular and alveolar edema. Histopathological analysis confirmed a significant reduction of tissue alteration, immune cell infiltration, and edema in sCPD9-ΔFCS- and mRNA-vaccinated animals (Fig. 3c–f). Inflammatory damage was milder in all animals infected with Omicron BA.5, and most pronounced in experimentally infected shedder hamsters (Fig. 3b and

 

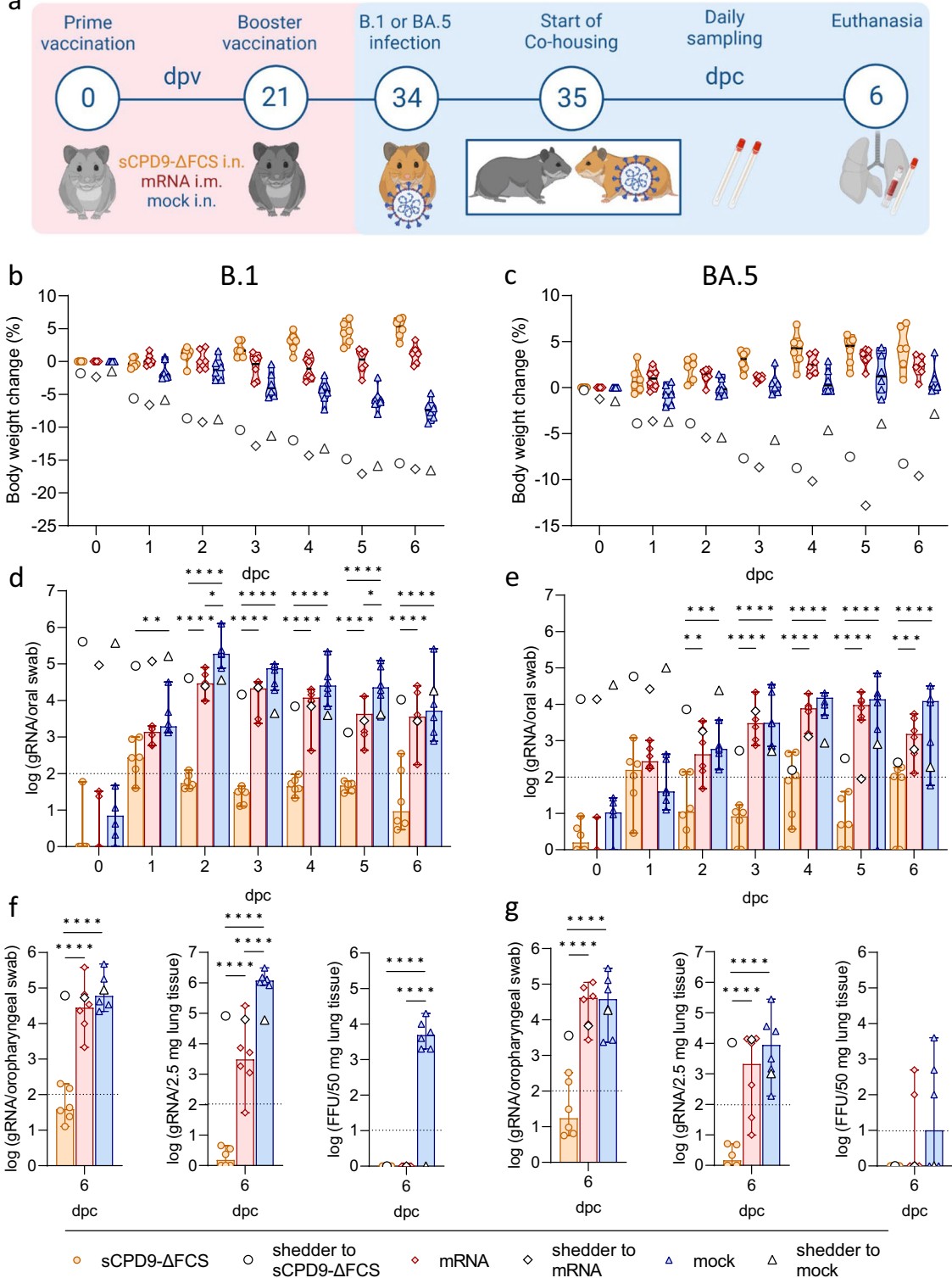

**Fig. 1 | Study scheme and virological results of vaccinated contact hamsters.**
**a** Schematic overview of the experiment. Syrian hamsters were vaccinated in a prime-boost setting with either sCPD9-ΔFCS intranasally (i.n.), an mRNA vaccine intramuscularly (i.m.), or a mock vaccine on days 0 and 21. Thirty-five days post vaccination (dpv) vaccinees were co-housed with SARS-CoV-2 B.1 or Omicron BA.5-infected shedder animals to screen for host-to-host transmission. After 6 days of contact (dpc), hamsters were euthanized to collect swabs, blood, and lung samples. Body weight changes of **b** B.1 shedder and contact animals, as well as **c** BA.5 shedder and contact animals. Violin plots (truncated) show weights of vaccinated contacts ($n = 6$), group medians, and quartiles. The weights of shedders ($n = 3$) are displayed as a median. Viral gRNA copies in oral swabs collected daily from **d** B.1 and **e** BA.5-infected shedder hamsters and their respective vaccinated contact animals. Viral gRNA copies in oropharyngeal swabs and lung tissue, and replicating virus quantified as focus forming units (FFU) of **f** B.1 and **g** BA.5-infected shedders and their respective vaccinated contacts. **d**–**g** Results of vaccinated contact animals ($n = 6$) are displayed as median with range, with symbols indicating individual values. For shedder hamsters ($n = 3$), medians are shown. **d**–**g** Parametric statistics on log-transformed data. **d**, **e** Ordinary two-way ANOVA with Tukey's multiple comparisons test was performed. **f**, **g** Ordinary one-way ANOVA with Tukey's multiple comparisons test was conducted. *$p < 0.05$, **$p < 0.01$, ***$p < 0.001$, and ****$p < 0.0001$. Source data are provided as a Source Data file.

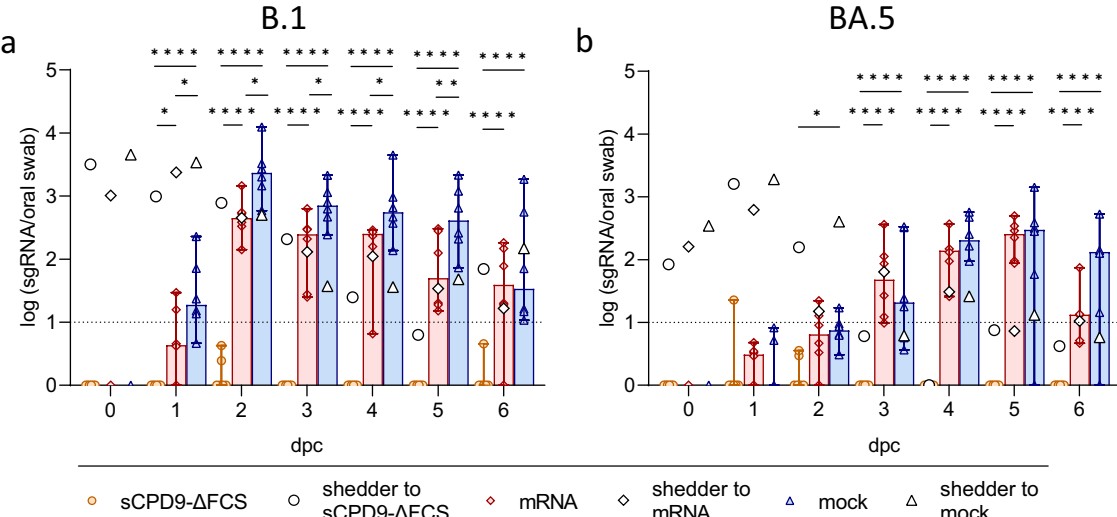

**Fig. 2 | Subgenomic RNA levels of vaccinated contact animals.** Viral sgRNA copy numbers in oral swabs collected daily from **a** B.1 and **b** BA.5-infected shedder hamsters and their respective vaccinated contact animals. **a, b** Results of vaccinated contact animals ($n = 6$) are displayed as median with range with symbols indicating individual values. For shedder hamsters ($n = 3$), medians are shown. Parametric statistics on log-transformed data. Ordinary two-way ANOVA with Tukey's multiple comparisons test was performed. $*p < 0.05$, $**p < 0.01$, $***p < 0.001$, and $****p < 0.0001$. Source data are provided as a Source Data file.

S2a, b). However, mock-vaccinated contacts still developed mild pneumonia (Fig. 3b). Protection was slightly less effective in mRNA-vaccinated hamsters. In contrast, sCPD9-ΔFCS-vaccinated animals exposed to B.1 or BA.5 shedders failed to develop substantial evidence of pneumonia, confirming the highly effective protection provided by this vaccine (Fig. 3).

**Strong humoral immune response accompanies the protective efficacy of the LAV**

To assess humoral immunity, the neutralizing capacity of sera collected at 6 dpc was evaluated against SARS-CoV-2 variants B.1, Delta, BA.1, and BA.5. In addition, enzyme-linked immunosorbent assays (ELISA) were conducted using sera and nasal washes collected at 6 dpc.

In agreement with previous studies[8–10], vaccination with sCPD9-ΔFCS elicited robust production of neutralizing antibodies. sCPD9-vaccinated hamsters produced comparable levels of neutralizing antibodies regardless of the virus variant used for the challenge (Fig. 4a). This finding, in conjunction with the virological results, suggests that mucosal immunity induced by sCPD9-ΔFCS vaccination prevented replication of the naturally transmitted virus. However, contact with virus antigens during co-housing with infected shedders may have resulted in a subtle increase in neutralizing activity. In contrast, mock-vaccinated animals displayed increased serum-neutralization capacity against the specific challenge virus. A similar trend was observed in mRNA-vaccinated animals, although the differences were less pronounced (Fig. 4a). As expected, specific serum-neutralization capacity was similar in all shedder animals and was determined by the challenge virus (Fig. S3a).

ELISAs revealed comparable levels of anti-B.1 spike and anti-BA.5 spike IgG antibodies in sCPD9-ΔFCS- and mRNA-vaccinated hamsters (Fig. 4b). As anticipated, nucleocapsid- and ORF3a-specific antibodies were solely present in sCPD9-ΔFCS-vaccinated animals, highlighting the broad immunity provided by LAVs (Fig. 4b). Not surprisingly, IgG levels in shedder animals were relatively uniform and strongly influenced by the challenge virus (Fig. S3b).

Mucosal immunity was investigated by measuring SARS-CoV-2-specific IgA levels in nasal washes obtained on 6 dpc (Fig. 4c). Irrespective of the challenge virus, sCPD9-ΔFCS-vaccinated animals showed similar levels of anti-B.1 spike and anti-BA.5 spike IgA antibodies (Fig. 4c). In contrast, only mRNA-vaccinated hamsters that were

exposed to B.1 shedders produced appreciable IgA levels. mRNA-vaccinated animals exposed to BA.5 shedders and both mock-vaccinated groups lacked measurable mucosal IgA response. The absence of IgA in nasal washes of mRNA-vaccinated hamsters that were in contact with BA.5 shedders confirms that mRNA vaccination confers only limited mucosal immunity before virus exposure. However, exposure to the homologous B.1 variant caused significant induction of mucosal IgA antibodies (Fig. 4c). Shedder animals exhibited low IgA levels, which corresponded to the virus used for infection (Fig. S3c).

**Vaccination with LAV prevents mucosal infection with SARS-CoV-2 B.1 and BA.5**

To further assess the protection in the upper airways induced by vaccination, nasal epithelium at 6 dpc was immunohistochemically evaluated for the presence of SARS-CoV-2 nucleocapsid antigen and histologically examined for signs of infection and inflammation. While SARS-CoV-2-positive cells were absent in nasal respiratory and olfactory epithelium of sCPD9-ΔFCS-vaccinated contacts, antigen was detected in abundance in mRNA- and mock-vaccinated animals (Fig. 4d, e).

In line with these results, the influx of immune cells and inflammatory damage were observed exclusively in the olfactory and respiratory epithelium of mRNA- and mock-vaccinated animals, with reduced inflammation in mRNA-vaccinated hamsters (Fig. 4d–f). When compared to B.1 shedders, inflammatory damage was less pronounced in animals exposed to BA.5 shedders. Consistent with previous observations[18–21], SARS-CoV-2[1818] antigen was markedly decreased or absent in shedder hamsters by 7 dpi. A low abundance of SARS-CoV-2 antigen was accompanied by mild signs of inflammation (Figure S3d).

**Vaccination with LAV blocks transmission of SARS-CoV-2 B.1 to naive contacts**

In the second set of experiments, we examined the effect of vaccination on limiting virus transmission from vaccinated and experimentally infected hamsters to naive animals. Syrian hamsters received two doses of sCPD9-ΔFCS or BNT162b2, administered 21 days apart. After 14 days, the hamsters were infected with either SARS-CoV-2 B.1 or Omicron BA.5. Twenty-four hours after infection, the infected animals were co-housed with naive contacts for 6 days, while monitoring their clinical status and body weight. Oral swabs were collected daily, and

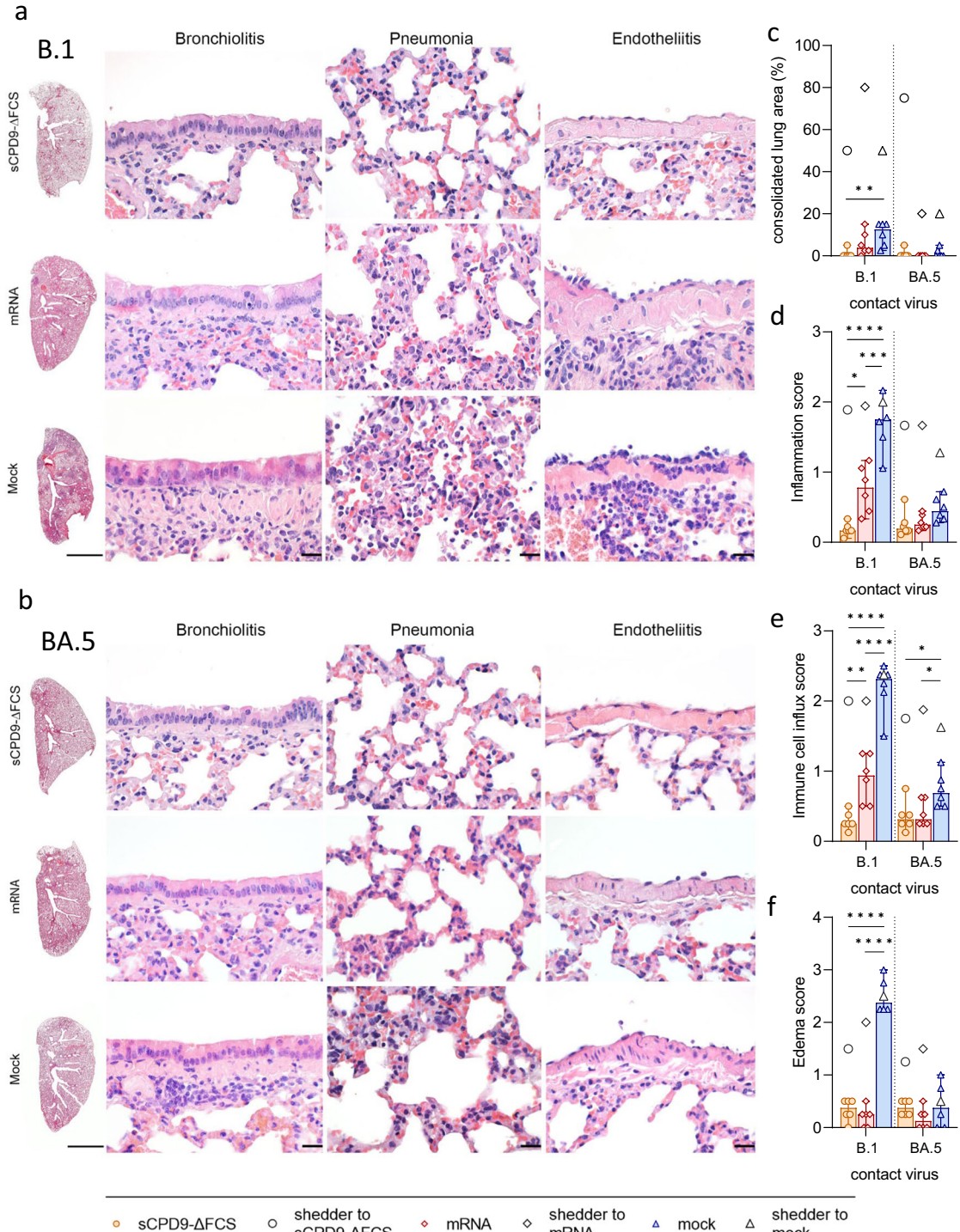

**Fig. 3 | Histopathological findings in vaccinated contact hamsters.**
**a**, **b** Overview images of hematoxylin and eosin-stained sections of the left lung lobes, and histopathology of bronchioli, alveoli, and vascular endothelium representing different manifestations of bronchiolitis, pneumonia, and endotheliitis in vaccinated animals in contact with **a** B.1 or **b** BA.5 shedders. Scale bars: left column 3 mm, all others 20 μm. **c**–**f** Semi-quantitative evaluation of pathological changes found in vaccinated hamsters ($n = 6$) in contact with B.1- or BA.5-infected shedders ($n = 3$). **c** Consolidated lung area in percentage. **d** Lung inflammation score accounting for the influx of neutrophils, lymphocytes and macrophages, bronchial epithelial necrosis, bronchitis, alveolar epithelial necrosis, perivascular lymphocyte cuffs as well as pneumocyte type II hyperplasia. **e** Immune cell influx score including infiltration of lymphocytes, neutrophils, and macrophages, as well as perivascular lymphocyte cuffs. **f** Pulmonary edema score accounting for perivascular and alveolar edema. **c**–**f** Values of contacts are displayed in median and range with symbols representing individual values. Results of shedder animals are shown as the median. Ordinary one-way ANOVA with Tukey's multiple comparisons test. *$p < 0.05$, **$p < 0.01$, ***$p < 0.001$, and ****$p < 0.0001$. Source data are provided as a Source Data file.

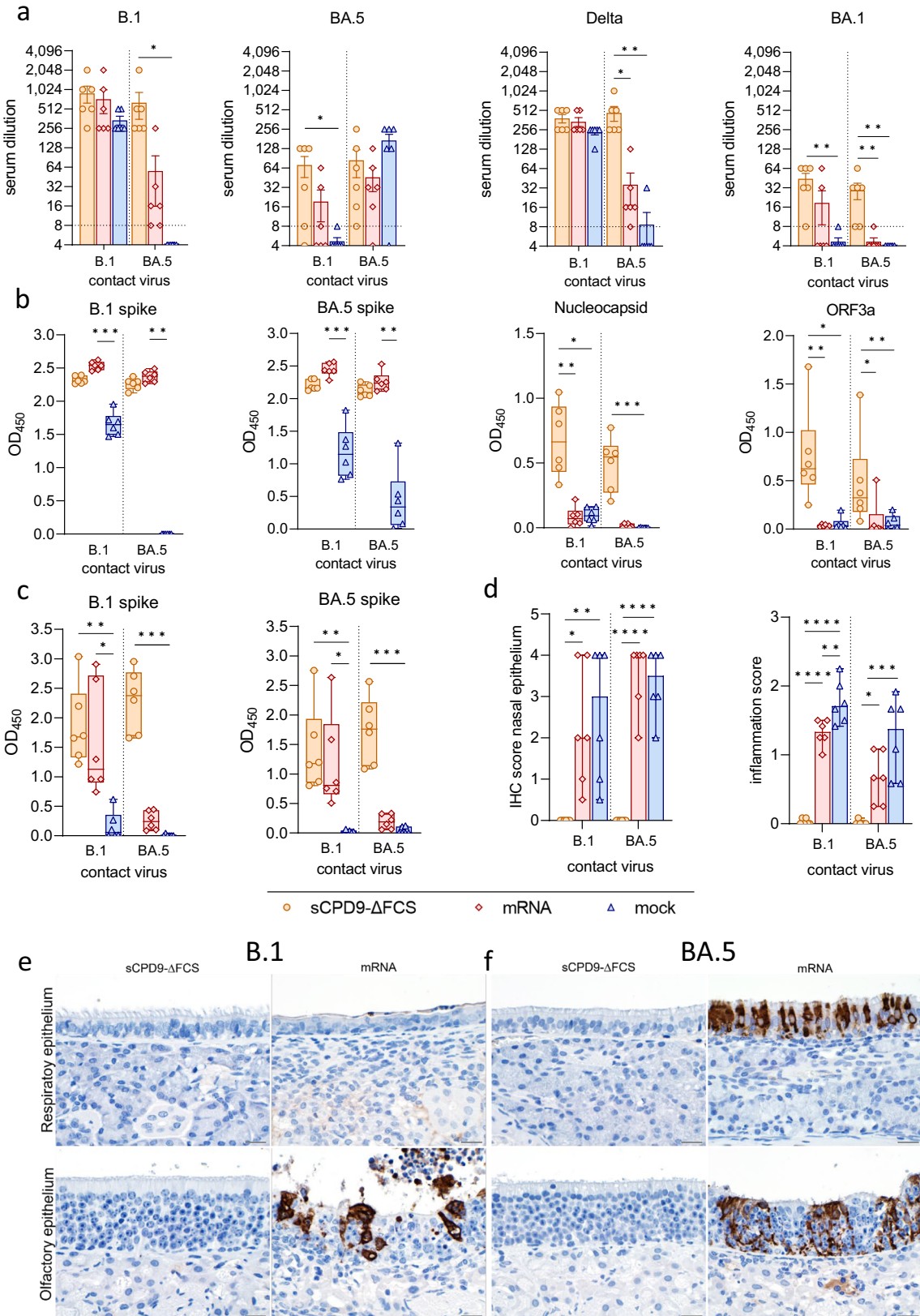

oropharyngeal swabs and lung samples were obtained on 6 dpc (Fig. 5a).

Both vaccines efficiently prevented body weight loss, whereas mock-vaccinated animals exhibited decreasing body weights upon B.1-infection (Fig. 5b and S4a). Virological and histopathological results confirmed the strong protective efficacy of the sCPD9-ΔFCS vaccine

upon SARS-CoV-2 B.1 challenge infection (Fig. S4b–f). Naïve contacts co-housed with sCPD9-ΔFCS-vaccinated and B.1-infected shedders maintained stable body weights. In contrast, contacts of mock- or mRNA-vaccinated and SARS-CoV-2-infected shedders experienced weight loss starting from 2 and 4 dpc, respectively (Fig. 5b), and their oral swabs exhibited high levels of SARS-CoV-2 gRNA (Fig. 5c and

**Fig. 4 | Systemic and mucosal immunity of vaccinated contact animals.**
**a** Neutralizing capacity against SARS-CoV-2 variants B.1, BA.5, Delta, and BA.1 of hamster sera taken 6 dpc from vaccinated animals (*n* = 6) in contact with either B.1 or BA.5 shedders (upper limit of detection = 1:1,024, lower limit of detection is indicated by dotted lines). Results are shown in mean ± SEM with symbols representing individual values. **b** SARS-CoV-2 specific IgG levels against B.1 spike, BA.5 spike, nucleocapsid, and ORF3a in serum collected from vaccinated contacts on day 6 after contact (dpc). **c** SARS-CoV-2 specific IgA levels against B.1 and BA.5 spike in nasal washes obtained from vaccinated contacts 6 dpc. **b**, **c** Findings displayed as optical density (OD) read at 450 nm. Box plots show 25th to 75th percentiles with centerlines indicating medians and whiskers from minimum to maximum. Symbols represent individual values of vaccinated contacts (*n* = 6) per

group. **a–c** Kruskal–Wallis test with Dunn's multiple comparisons test. *\*p* < 0.05, *\*\*p* < 0.01, *\*\*\* p* < 0.001, and *\*\*\*\*p* < 0.0001. **d** Semi-quantitative scoring of SARS-CoV-2 N protein immunohistochemistry (IHC) signal in nasal epithelium of vaccinated contacts (*n* = 6). Scoring of inflammatory changes in nasal epithelium including an influx of lymphocytes, neutrophils, olfactory and respiratory epithelial necrosis, apoptosis, loss of cilia, and flattened epithelial cells displayed as median with range. Symbols indicate individual values. Ordinary one-way ANOVA with Tukey's multiple comparisons test. *\*p* < 0.05, *\*\*p* < 0.01, *\*\*\*p* < 0.001, and *\*\*\*\*p* < 0.0001. **e, f** Immunohistochemical staining for SARS-CoV-2 nucleocapsid antigen in nasal epithelial cells with 3,3'-Diaminobenzidine chromogen (brown) and hemalum counterstain (blue). Scale bars: 20 μm. Source data are provided as a Source Data file.

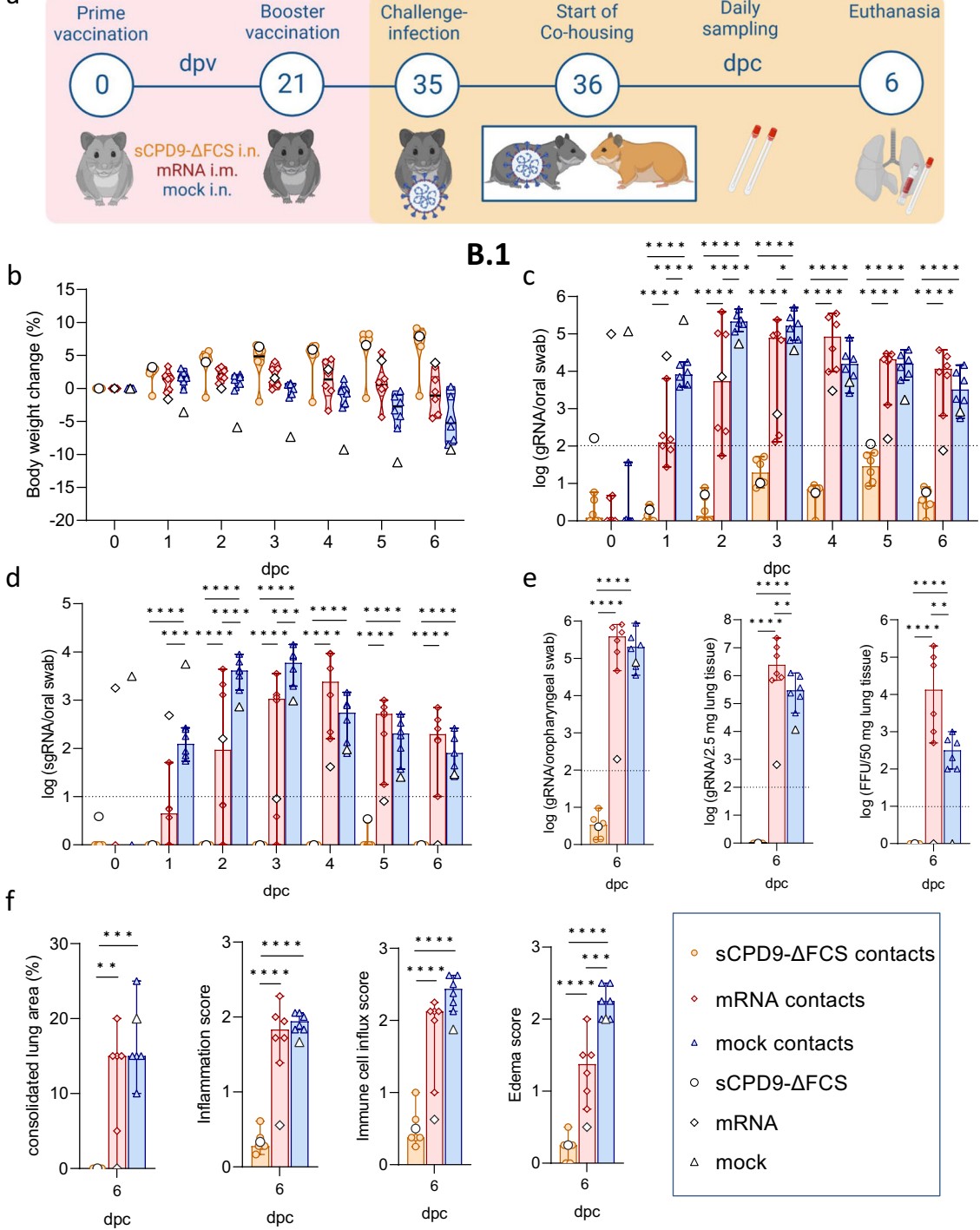

**Fig. 5 | Study scheme, clinical, virological, and histopathological results of naive contacts of vaccinated B.1 shedders. a** Study scheme. Following a prime-boost schedule, Syrian hamsters received either sCPD9-ΔFCS intranasally (i.n.), an mRNA vaccine intramuscularly (i.m.), or a mock vaccine on days 0 and 21. Subsequently, vaccinated animals were challenge-infected with SARS-CoV-2 variants B.1 or Omicron BA.5, 35 days after receiving prime vaccination (dpv). Twenty-four hours after challenge infection, challenge-infected animals were cohabitated with immunologically naive animals to assess host-to-host transmission. Hamsters were euthanized 6 days post contact (dpc) to collect swabs, blood, and lung samples. **b** Body weight loss in percentage from vaccinated and B.1-infected shedders and naive contact animals. Violin plots (truncated) represent weights of naive contacts ($n = 6$) in group medians and quartiles. The weights of shedders ($n = 3$) are shown as the median. Viral **c** gRNA copies and **d** sgRNA copies in oral swabs from vaccinated and B.1-infected shedder hamsters and naive contacts. **e** Viral gRNA copies in oropharyngeal swabs and 2.5 mg homogenized lung tissue collected at

termination. Replicating virus in homogenized lung tissue quantified as focus forming units (FFU). **f** Histopathological scoring of hamster lungs displaying the percentage of lung consolidation, inflammatory damages including influx of neutrophils, lymphocytes and macrophages, bronchial epithelial necrosis, bronchitis, alveolar epithelial necrosis, perivascular lymphocyte cuffs as well as pneumocyte type II hyperplasia. Immune cell influx score accounts for lymphocyte, neutrophil, and macrophage infiltration, as well as perivascular lymphocyte cuffs. Edema score includes perivascular and alveolar edema. **c–f** Results of naive contact animals ($n = 6$) are shown in median with range. Symbols represent individual values. Results of vaccinated and infected hamsters ($n = 3$) are displayed as the median. **c–e** Parametric statistics on log-transformed data. The dotted line shows the limit of detection. **c, d** Ordinary two-way ANOVA with Tukey's multiple comparisons test. **e, f** Ordinary one-way ANOVA with Tukey's multiple comparisons test. *$p < 0.05$, **$p < 0.01$, ***$p < 0.001$, and ****$p < 0.0001$. Source data are provided as a Source Data file.

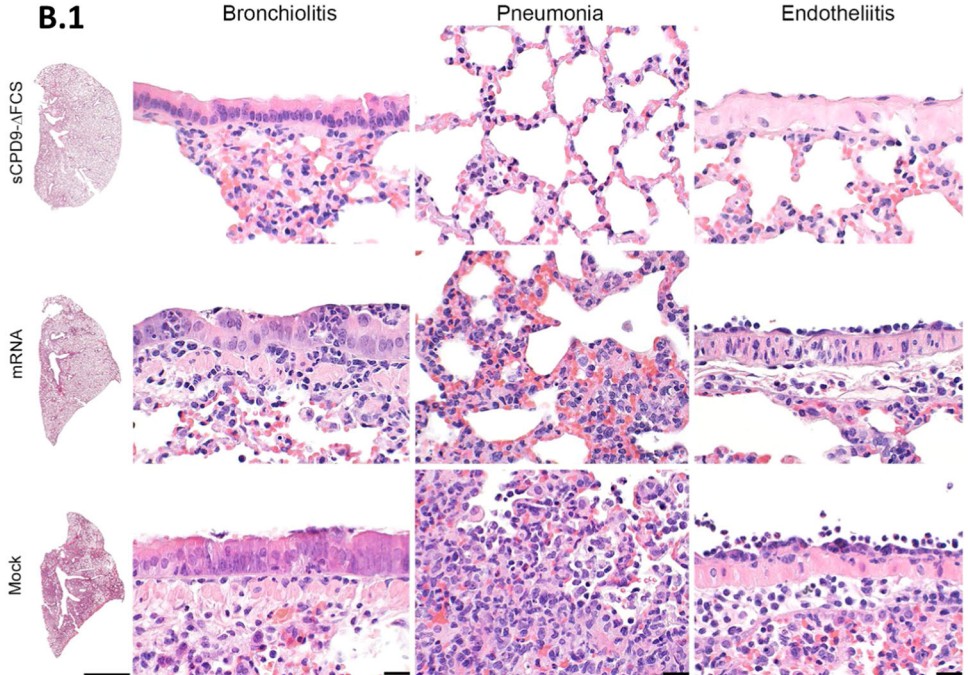

**Fig. 6 | Histopathological findings in naive contacts of vaccinated SARS-CoV-2 B.1 shedders.** Representative histopathology of $n = 3$ hamsters per group as indicated displaying different severities of bronchiolitis, pneumonia, and endothelialitis. Scale bars: left column 3 mm, all others 20 μm.

Table S3). Moreover, sgRNA, which is produced during active virus replication, was detected in oral swabs of the latter groups (Fig. 5d, and Table S4). However, contacts of mRNA-vaccinated shedders showed slightly delayed virus replication kinetics compared to contacts of mock-vaccinated animals (Fig. 5c, d). The presence of high gRNA levels in lungs, and the detection of replication-competent virus further confirmed that all naive animals in contact with mRNA- or mock-vaccinated and B.1-infected hamsters became infected with the challenge virus (Fig. 5e). In contrast, contacts of sCPD9-ΔFCS-vaccinated and B.1-infected animals remained negative for SARS-CoV-2 sgRNA and gRNA in the upper and lower airways, with no replicating virus detected in the lower airways. These findings strongly indicate that vaccination with sCPD9-ΔFCS confers highly effective protection against onward transmission of the B.1 virus to naive contacts. (Fig. 5c–e).

Consistent with these observations, hamsters that contracted B.1 infection from vaccinated and challenged animals showed typical signs of COVID-19 pneumonia, including necrosuppurative bronchitis and bronchiolitis, the proliferation of alveolar type II epithelia, vascular endotheliitis, diffuse alveolar damage as well as perivascular and alveolar edema. However, the influx of immune cells and the edema

were reduced in hamsters co-housed with mRNA-vaccinated animals compared to mock-vaccinated counterparts. Importantly, none of the hamsters in contact with sCPD9-ΔFCS-vaccinated and B.1-infected animals showed signs of pneumonia (Figs. 5f and 6).

## LAV provides superior protection against Omicron BA.5 infection and onward transmission

Owing to its attenuation in Syrian hamsters[14,22] no significant weight loss was observed in hamsters infected with Omicron BA.5 (Fig. 7a and S5a). In animals vaccinated with sCPD9-ΔFCS and subsequently infected with Omicron BA.5, gRNA copy numbers rapidly decreased and were below the detection limit 2 dpc. In contrast, mRNA vaccination only marginally reduced the viral load in the upper and lower respiratory tract when compared to mock vaccination. Importantly, SARS-CoV-2 gRNA was detected in oral swabs of mRNA-vaccinated hamsters until 5 dpc (Fig. 7b, c and S5b, c). Consistent with these results, sgRNA was only detectable on the day directly following challenge virus inoculation in sCPD9-ΔFCS-vaccinated hamsters. In contrast, the presence of sgRNA suggests replication of BA.5 for up to 5 days in mRNA- and up to 7 days in mock-vaccinated animals, enabling onward transmission in all cases (Fig. 7c and S5d). Histopathological

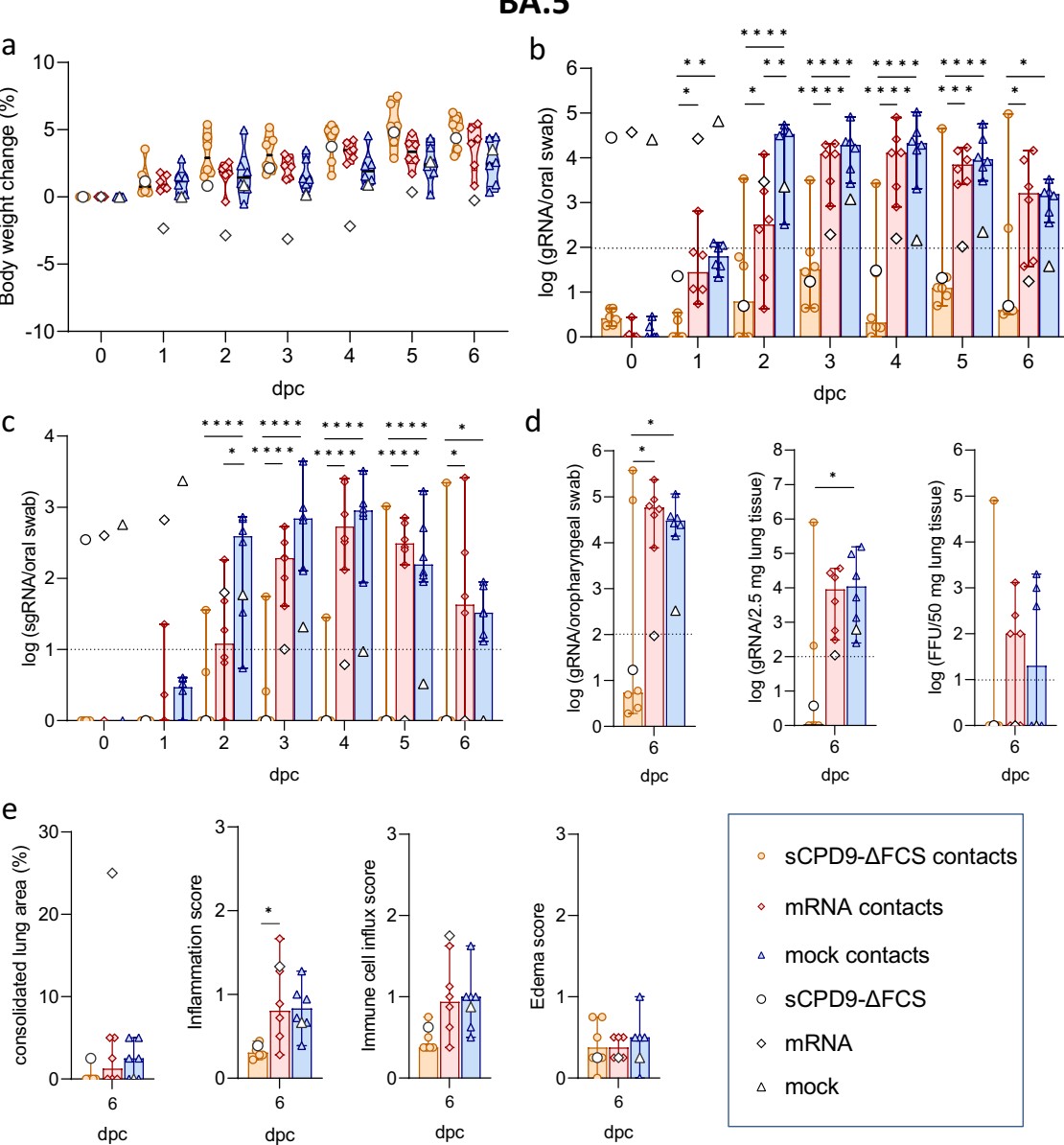

**Fig. 7 | Clinical, virological, and histopathological results of naive contacts of vaccinated BA.5 shedders. a** Body weight loss in percent of vaccinated and BA.5-infected shedder animals and naive contacts. Violin plots (truncated) represent weights of naive contacts ($n = 6$) in group medians and quartiles. The weights of shedders ($n = 3$) are shown as median. Viral **b** gRNA copies and **c** sgRNA copy numbers in daily oral swabs of vaccinated and challenge-infected BA.5 shedders and naive contact hamsters. **d** gRNA copy numbers detected in oropharyngeal swabs and 2.5 mg lung tissue obtained at termination. Replication-competent virus in 50 mg homogenized lung shown as focus forming units (FFU).
**e** Histopathological scoring of hamster lungs showing the percentage of lung consolidation. Pneumonia score includes the influx of neutrophils, lymphocytes, and macrophages, bronchial epithelial necrosis, bronchitis, alveolar epithelial necrosis, perivascular lymphocyte cuffs as well as pneumocyte type II hyperplasia. Immune cell influx score accounting for lymphocyte, neutrophil, and macrophage infiltration, as well as perivascular lymphocyte cuffs. Edema score including perivascular and alveolar edema. **b–e** Results of naive contact animals ($n = 6$) are displayed as median with range with symbols representing individual values. For findings in vaccinated and challenge-infected shedders ($n = 3$), the median value is shown. **b–d** Parametric statistics on log-transformed data. The dotted line represents the limit of detection. **b, c** Ordinary two-way ANOVA with Tukey's multiple comparisons test. **d, e** Ordinary one-way ANOVA with Tukey's multiple comparisons test. *$p < 0.05$, **$p < 0.01$, ***$p < 0.001$, and ****$p < 0.0001$. Source data are provided as a Source Data file.

examination of vaccinated- and B.1-infected hamsters confirmed the efficacy of both vaccines, although the histopathological changes observed in BA.5-infected animals were generally more subtle compared to changes observed in B.1-infected animals, suggesting an overall milder pathology with this variant compared to the primordial virus (Fig. S5e, f).

No weight loss was observed in naive groups in contact with vaccinated and BA.5-infected hamsters (Fig. 7a). However, mRNA vaccination failed to prevent onward transmission of the BA.5 virus.

Starting from 3 dpc, all animals in contact with mRNA- or mock-vaccinated shedders had comparable SARS-CoV-2 gRNA and sgRNA loads in the upper respiratory tract (Fig. 7b, c). Additionally, high SARS-CoV-2 gRNA levels were also found in oropharyngeal swabs and lung samples collected from these animals at termination (Fig. 7d). In contrast, sCPD9-ΔFCS vaccination greatly reduced transmission to naive contacts, with only one animal contracting the infection around 2 dpc. A second naive animal in the same cage tested positive at 6 dpc, indicating secondary transmission (Fig. 7b). Both hamsters tested

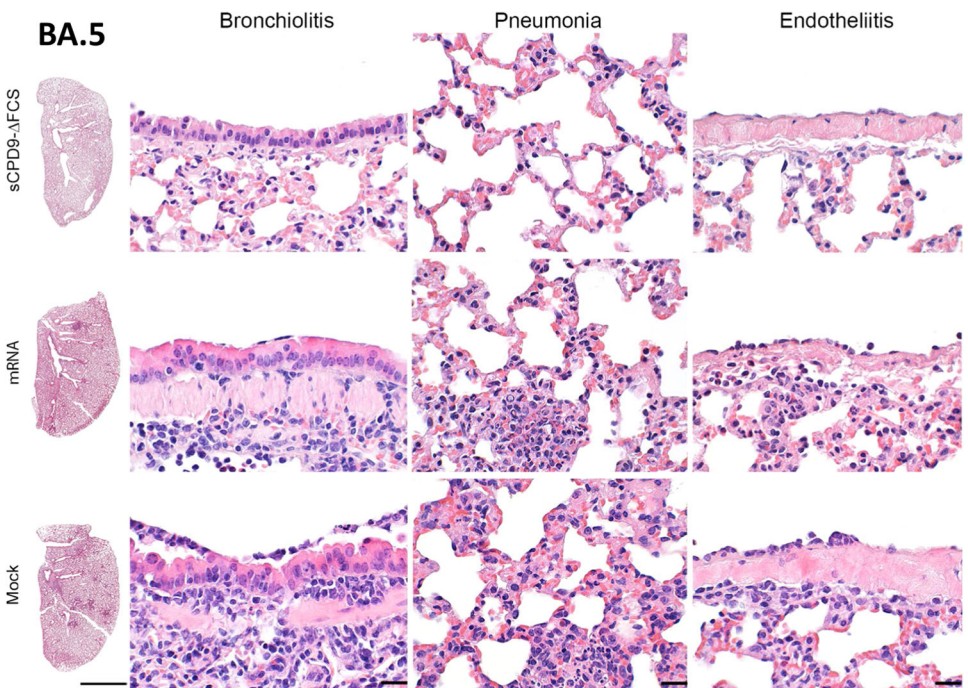

**Fig. 8 | Histopathological findings in naive contacts of vaccinated Omicron BA.5 shedders.** Representative histopathology of $n = 3$ hamsters per group as indicated displaying different severities of bronchiolitis, pneumonia, and endothelialitis. Scale bars: left column 3 mm, all others 20 μm.

positive for SARS-CoV-2 gRNA in oropharyngeal swabs and lungs at termination, but only one animal had replicating virus at termination. Meanwhile, the replicating virus was present in three contacts of both mRNA- and mock-vaccinated shedders (Fig. 7d).

Histopathological findings were less pronounced in contacts exposed to vaccinated and BA.5-infected hamsters. Contacts of mock- or mRNA-vaccinated hamsters showed mild to moderate pneumonia with an increased influx of immune cells, but no lung consolidation. Naive contacts of sCPD9-ΔFCS-vaccinated animals showed either no signs or mild lung inflammation, reflecting their infectious status (Figs. 7e and 8).

### Humoral and mucosal immunity induced by LAV reduced onward transmission of SARS-CoV-2

Naive contacts of sCPD9-ΔFCS-vaccinated and B.1-infected shedders showed no seroconversion, while contacts of mRNA- or mock-vaccinated and infected shedders exhibited seroconversion dependent on the challenge virus (Fig. 9a, b). As expected, sCPD9-ΔFCS-vaccinated animals exhibited broad and strong humoral immune responses prior to infection, while mRNA-vaccinated hamsters directed their humoral response solely against the B.1 spike protein (Fig. S6a–d). Neutralizing antibody titers against BA.5 were only detected in animals that were vaccinated with sCPD9-ΔFCS (Fig. S6a, d). The challenge infection boosted the antibody response in all groups (Fig. S6c, d).

In agreement with virological results, only one naive contact of an sCPD9-ΔFCS-vaccinated and BA.5-infected hamster showed seroconversion. No antibodies against SARS-CoV-2 B.1, Delta, and BA.1 were detected in any of the serum samples obtained from contact animals of BA.5 shedders. Moreover, antibodies directed against B.1 proteins S, N, and ORF3a were not detectable by ELISA (Fig. 9b). Overall, only sCPD9-ΔFCS vaccination induced broad humoral immunity and effectively reduced BA.5 transmission to naive contacts.

Nasal washes of sCPD9-ΔFCS-vaccinated and infected shedders showed high IgA levels against B.1 and Omicron BA.5 spikes, irrespective of the challenge virus. Meanwhile, mRNA- and mock-vaccinated and infected hamsters had low or no IgA levels,

confirming the superior mucosal immunity provided by intranasal vaccination (Fig. S7a). Naive contacts had no IgA antibodies on 6 dpc, but there was a minor tendency towards IgA development in contacts of mRNA- and mock-vaccinated and infected animals, aligning with virological and serological findings (Fig. 9c).

Naive contacts of mRNA-vaccinated shedders contracted both SARS-CoV-2 variants and showed abundant expression of the nucleocapsid in the nasal epithelium (Fig. 9d–f). Contacts of mock-vaccinated and B.1-challenged hamsters had fewer SARS-CoV-2-positive cells in nasal epithelium compared to contacts of mRNA-vaccinated animals, consistent with virus RNA levels in oral swabs on 6 dpc (Fig. 9d).

In line with previous observations, nasal epithelium of naive contacts of sCPD9-ΔFCS-vaccinated and B.1- or BA.5-infected shedders was free of SARS-CoV-2 nucleocapsid protein, except for the single hamster that contracted BA.5 infection (Fig. 9d). In accordance with immunohistochemical results, a variable degree of inflammation and immune cell recruitment was detected in all hamsters that contracted the infection with either of the two variants (Fig. 9d–f), corroborating the effectiveness of sCPD9-ΔFCS in preventing virus transmission. Histological findings confirmed complete clearance of the infection in all vaccinated and infected groups (Fig. S7b–d).

## Discussion

Existing COVID-19 vaccines have been successful in reducing hospitalizations and deaths, but constantly emerging SARS-CoV-2 variants compromise their effectiveness[23–25]. In particular, the highly transmissible Omicron variants have diminished the effectiveness of BA.5-adapted vaccines shortly after their introduction to the market[3,4,25]. In the pursuit of inducing mucosal immunity at the site of initial virus infection, the intranasal delivery route is certainly advantageous. This consideration prompted trials that assessed the effectiveness of mRNA and vector-based vaccines after intranasal administration. Although the vaccines triggered humoral immunity and reduced clinical symptoms and viral loads, they failed to prevent SARS-CoV-2 infection in preclinical models[26–28]. Moreover, systemic neutralizing antibodies induced by intranasal mRNA vaccination exhibited only limited cross-

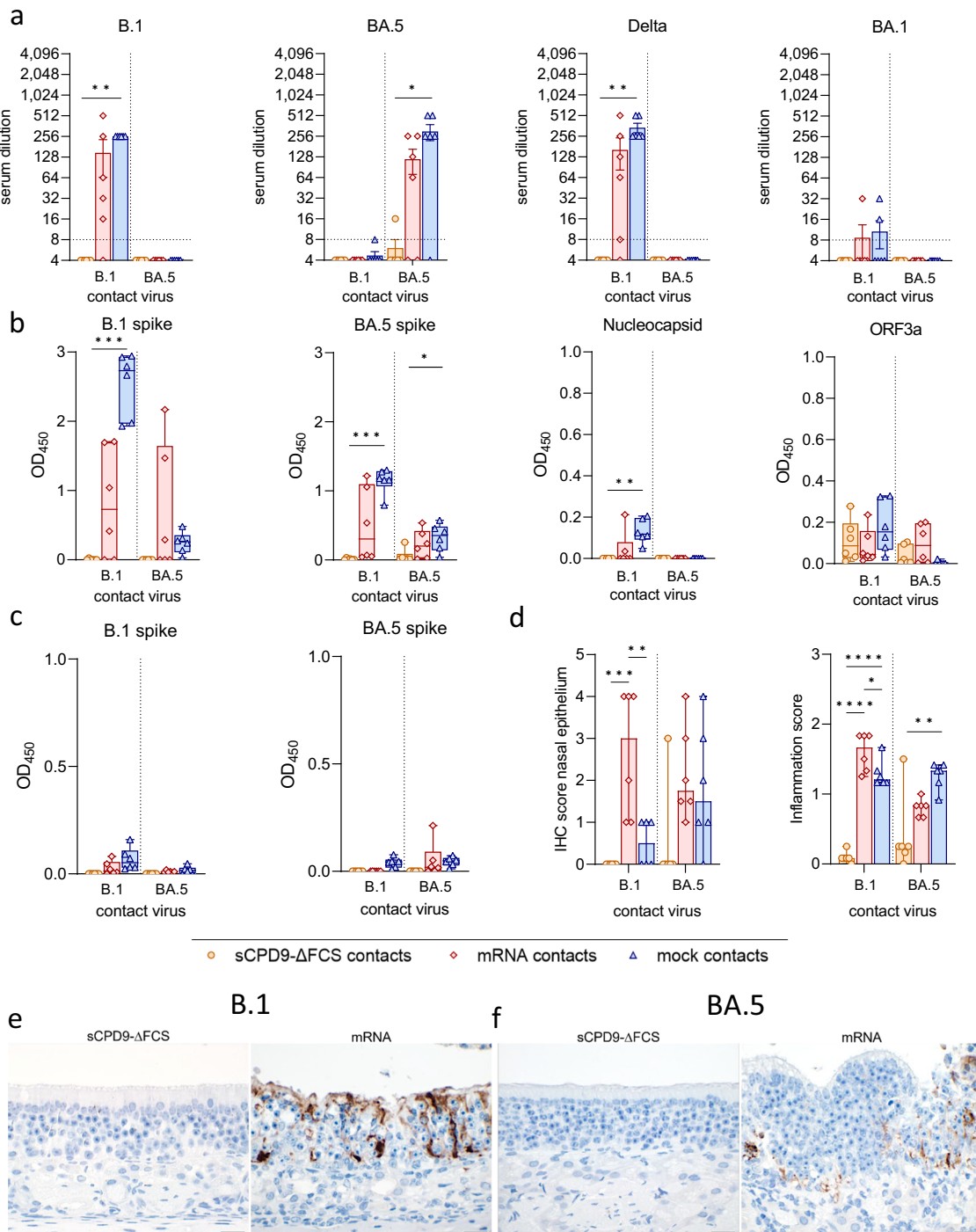

**Fig. 9 | Systemic and mucosal immunity of naive contact animals. a** Neutralizing antibodies against SARS-CoV-2 variants B.1, BA.5, Delta, and BA.1 in sera collected at 6 dpc from naive hamsters ($n = 6$) in contact with vaccinated and B.1 or BA.5-infected shedders. The lower limit of detection is indicated by dotted lines, the upper detection limit is 1:1,024. Results are displayed in mean ± SEM. **b** SARS-CoV-2 specific IgG levels against B.1 spike, BA.5 spike, nucleocapsid, and ORF3a in sera from naive contacts collected at 6 dpc. **c** SARS-CoV-2 specific IgA levels against B.1 spike and BA.5 spike in nasal washes obtained at termination. **b, c** Findings are displayed as optical density (OD) at 450 nm. Box plots show 25th to 75th percentiles with centerlines indicating medians, whiskers from minimum to maximum, and

individual values of naive contacts indicated by symbols ($n = 6$). **d** Semi-quantitative scoring of SARS-CoV-2 N protein immunohistochemistry (IHC) in nasal epithelium of naive contacts ($n = 6$). Scoring of inflammatory changes in nasal epithelium including influx of lymphocytes, neutrophils, olfactory and respiratory epithelial necrosis, apoptosis, loss of cilia, and flattened epithelial cells. Ordinary one-way ANOVA with Tukey's multiple comparisons test. **e, f** Immunohistochemical staining for SARS-CoV-2 nucleocapsid antigen in nasal epithelial cells with DAB chromogen (brown) and hemalum counterstain (blue). Scale bars: 20 μm. **a–d** Kruskal–Wallis test with Dunn's multiple comparisons test. *$p < 0.05$, **$p < 0.01$, ***$p < 0.001$, and ****$p < 0.0001$. Source data are provided as a Source Data file.

reactivity against emerging Omicron variants[28]. Consequently, there is a growing consensus that the development of new vaccines or formulations capable of eliciting strong and broad mucosal immunity is necessary[12,29,30].

Dimeric IgA antibodies are the dominant antibody type in mucosal tissues, particularly in the upper respiratory tract where SARS-CoV-2 is first encountered. A recent study has shown that monomeric IgA antibodies have lower neutralizing activity than

corresponding IgG monomers, but that dimeric IgA antibodies are 15 times more potent than their monomeric counterparts[31]. Consequently, vaccines that stimulate the production of dimeric IgA antibodies at mucosal surfaces may offer superior protection against SARS-CoV-2. While IgG is crucial in protective immunity in the lower respiratory tract, IgA is relatively more important in the upper respiratory compartment, the primary site of initial infection and viral shedding[32–36]. Previously, we showed that sCPD9 induces more potent IgA antibody responses than vaccines that are delivered intramuscularly[10]. This study further corroborates these findings, demonstrating that sCPD9-ΔFCS vaccination induces robust levels of IgA antibodies capable of effectively neutralizing the two evolutionary distant virus variants tested here.

The extreme antigenic plasticity of the SARS-CoV-2 spike protein makes spike-based vaccines highly susceptible to immune evasion through antigenic drift[37]. However, specifically T-cell immunity can target several more conserved antigens outside the spike protein, thereby significantly broadening anti-SARS-CoV-2 immunity, also across different virus variants[38]. Our LAV presents the virus's entire antigenic repertoire at the respiratory mucosa, which among other things, triggers the formation of tissue-resident memory T cells (TRM cells), a specialized subset of T cells that remain stationary in specific tissues, such as the respiratory mucosa, after an initial infection or vaccination[39,40]. These cells provide a first line of defense against reinfection, by rapidly recognizing and responding to pathogens that re-enter the tissue at body surfaces[40,41], and contribute to the early control of viral replication and limit virus spread within the respiratory tract. Together with neutralizing (IgA) antibodies, TRM cells contribute to a comprehensive defense against SARS-CoV-2, effectively targeting the virus at its entry point and initial replication site, thereby reducing the likelihood of respiratory tract infection and transmission[10].

In previous work, we demonstrated that sCPD9 offers superior protection against infection with SARS-CoV-2 compared to intramuscularly administered spike-based mRNA and vectored vaccines[10]. In this study, we aimed to compare the effectiveness of the intranasally applied attenuated virus sCPD9-ΔFCS and the intramuscularly injected mRNA vaccine BNT162b2 in controlling the transmission of SARS-CoV-2. Both vaccines encode the original form of the SARS-CoV-2 spike protein. We provide direct evidence demonstrating the superior capacity of the attenuated virus to prevent or significantly reduce virus transmission. Importantly, this remains true even for BA.5, an evolved, highly transmissible, and strongly immune evasive SARS-CoV-2 variant. The emergence of Omicron variants, which carry numerous amino acid changes in their spike protein[13,42], spurred the development of bivalent mRNA vaccines. These vaccines contain the spike protein of the B.1 variant, and of the BA.4/BA.5 variant, providing superior protection against Omicron variants compared to monovalent vaccines[2,43]. However, it becomes increasingly clear that SARS-CoV-2 transmission is not, or is not sufficiently controlled by intramuscular spike-based vaccines[44].

Given that the epithelium of the upper respiratory tract serves as the primary site of SARS-CoV-2 infection, the development of intranasal vaccines capable of eliciting robust and durable immunity is highly desirable. While current SARS-CoV-2 mRNA vaccines have not been designed for intranasal delivery, one study demonstrated their ability to provide limited protection against SARS-CoV-2 when administered intranasally in experimental models[28]. These findings suggest the feasibility of developing mRNA vaccines specifically tailored for intranasal administration. In addition, comparative studies evaluating the efficacy of various vaccine types delivered through the same route, such as intranasal administration, could offer valuable insights into their performance. However, optimizing mRNA vaccines for intranasal application may require substantial modifications of the vaccine formulation given the unique characteristics of mucosal delivery. Critical factors, such as vaccine formulation, adjuvants, and the target antigen must be carefully considered as they can significantly influence the outcomes. Since our study provides evidence for superior protection offered by an intranasal vaccine, comparing different intranasal vaccines and formulations will be an important area of future research.

Further, our findings indicate that, in the case of SARS-CoV-2, attenuated mucosal vaccines may have the ability to efficiently prevent or reduce virus infection and onward transmission. Additionally, they may not require frequent updates of the viral antigens, as our B.1-based vaccine provides highly efficient protection against the BA.5 variant that is antigenically far distant from the B.1 variant. Our research has also shown that the administration of two consecutive doses of sCPD9 effectively enhances immunity. This indicates that pre-existing immunity, such as that conferred by previous SARS-CoV-2 infection, does not impede the effectiveness of the mucosal vaccine tested here. On the contrary, periodic boosting of existing immunity through mucosal vaccines could serve as an important strategy for the long-term control of SARS-CoV-2. An important limitation of our study is that we did not investigate the durability of vaccine-induced protection. Future studies will address the effectiveness of vaccination over the course of several months, as well as optimal boosting regimens.

In conclusion, the findings presented here underscore the significance and benefits of developing mucosal vaccines to enhance control of not only SARS-CoV-2 but potentially also other respiratory viruses. Reducing virus transmission may constrain and slow respiratory RNA virus circulation and evolution.

## Methods

### Study design

In this study, we aimed to compare the protection against SARS-CoV-2 transmission provided by two different types of vaccines. The extent of SARS-CoV-2 transmission from infected shedder hamsters to vaccinated animals and from vaccinated and subsequently infected shedders to naive contact hamsters was investigated in two different trials. For this purpose, hamsters were randomly assigned into groups of 12 animals. The experimental design of both studies included three groups following a prime-boost schedule in which the hamsters were vaccinated on day 0 and day 21 respectively. The three groups received either two doses of sCPD9-ΔFCS ($10^4$ FFU) intranasally, 5 μg of the mRNA vaccine BNT162b2 (Comirnaty®, Pfizer–BioNTech) intramuscularly, or two doses of mock vaccine intranasally.

In the first trial, two vaccinated hamsters were co-housed with one infected shedder animal per cage, commencing 2 weeks after receiving the booster vaccination. Twenty-four hours prior to co-habitation, the shedder hamsters were infected with $10^5$ FFU of either SARS-CoV-2 variant B.1 or the Omicron subvariant BA.5 by intranasal instillation, performed under general anesthesia. For the second experiment, 2 weeks after the second vaccination, the vaccinated animals were anesthetized to allow blood collection and infection. Following blood collection, they were challenge-infected intranasally with $10^5$ FFU of either SARS-CoV-2 variant B.1 or Omicron subvariant BA.5. Twenty-four hours after infection, one vaccinated and infected animal was co-housed with two naive contact hamsters per cage. In both trials, hamsters were co-housed for 6 days. Throughout this period, oral mucosal swabs were taken daily to determine viral loads in the upper airways of shedder and contact hamsters. In addition, swabs were taken from contact animals, on day 0 post contact–prior to the contact with shedder animals. Body weights and clinical conditions were assessed daily. On day 6 after contact, all hamsters were euthanized, and blood, nasal washes, oropharyngeal swabs, lungs, and skulls were collected for subsequent virological, serological, and histopathological examinations.

### Cells

Vero E6 (ATCC CRL-1586) and Vero E6-TMPRSS2 (NIBSC 100978) cells were cultivated in minimal essential medium (MEM) containing 10%

fetal bovine serum (PAN Biotech), 100 IU/ml penicillin G and 100 mg/ml streptomycin (Carl Roth). To ensure the selection of TMPRSS2-expressing cells, the medium for Vero E6-TMPRSS2 cells contained an additional 1000 µg/ml geneticin (G418). CaLu-3 cells (ATCC HTB-55) were grown in Dulbecco's Modified Eagle Medium (DMEM, Gibco) with 20% fetal bovine serum (PAN Biotech), 1% non-essential amino acids, 100 IU/ml penicillin G and 100 mg/ml streptomycin (Carl Roth). All cells were cultivated at 37 °C and 5% $CO_2$.

## Viruses
The SARS-CoV-2 variant B.1 (B.1, hCoV-19/Germany/BY-ChVir-929/2020, EPI_ISL_406862) propagated on Vero E6 cells and the Omicron subvariant BA.5 (BE.1.1, hCoV-19/Germany/SH-ChVir29057_V34/2022, EPI_ISL_16221625) grown on CaLu-3 cells, were used to infect the shedder hamsters. Additionally, Delta variant B.1.617.2 (B.1.617.2, Human, 2021, Germany ex India, 20 A/452 R, EVAg: 009V-04187) propagated on Vero E6-TMPRSS2 cells and Omicron subvariant BA.1 (BA.1.18, hCoV-19/Germany/BE-ChVir26335/2021, EPI_ISL_7019047) grown on CaLu-3 cells were used to conduct serum-neutralization assays. Plaque assays were performed on Vero E6 cells to determine the titer of all virus stocks prior to the infection experiment. Vials were stored at −80 °C.

## Ethics statement
Animal works were performed in compliance with all applicable national and international regulations and approved by the regulatory state authority, Landesamt für Gesundheit und Soziales in Berlin, Germany (permit number 0086/20). All in vitro and animal experiments were conducted in the certified BSL-3 laboratory at the Institut für Virologie, Freie Universität Berlin, Berlin, Germany.

## Animal husbandry
Male Syrian hamsters (*Mesocricetus auratus;* breed RjHan:AURA) were purchased from Janvier Labs at 5–7 weeks of age. They were kept in groups of 2 to 3 animals in individually ventilated cages (IVCs; Tecniplast) equipped with nesting material. Prior to vaccination, the animals were allowed to habituate to the housing conditions for seven days. The hamsters had free access to water and food at all times. During all experiments, cage temperature and relative humidity were monitored and ranged between 22 to 24 °C and 40 and 55%.

## Vaccine preparation and vaccination
The live-attenuated vaccine candidate sCPD9-ΔFCS was propagated on Vero E6-TMPRSS2 cells. Titers were determined by plaque assays conducted on Vero E6 cells. Prior to vaccination, the stock was adjusted to a final titer of $2 \times 10^5$ FFU/ml. Intranasal vaccination with $10^4$ FFU per animal was performed under general anesthesia (0.15 mg/kg medetomidine, 2.0 mg/kg midazolam, and 2.5 mg/kg butorphanol).

BNT162b2 (Comirnaty®) was prepared according to the manufacturer's instructions. The final concentration of mRNA was diluted to 50 µg/ml instead of 100 µg/ml as recommended for use in humans. The dilution was prepared with 0.9% NaCl sterile water immediately prior to vaccination and applied intramuscularly at a dose of 5 µg per hamster.

Animals assigned to the mock group received minimal essential medium (MEM) intranasally under general anesthesia.

## Nasal washes
To obtain nasal washes, the skulls paramedian of the nasal septum were punctured with a cannula. Subsequently, a pipette tip was inserted and 200 µl of PBS was gently injected into the nasal cavity. The wash fluid was collected from the nostrils, and the washing procedure was repeated twice. Approximately 150 µl of nasal wash fluid was collected per animal.

## Swab sampling
Oral swabs were collected daily from all hamsters to monitor viral gRNA and sgRNA loads throughout the experiment. However, at termination (6 dpc), an oropharyngeal swab was collected in addition to the daily oral swab for increased sensitivity in the detection of viral RNA in the upper respiratory tract.

## RNA isolation and RT-qPCR
To extract RNA from lung tissue, 25 mg of lung tissue was first homogenized in a bead mill (Analytic Jena). RNA was isolated from oral swabs, oropharyngeal swabs, and the homogenized lung tissue using the innuPREP Virus DNA/RNA Kit (Analytik Jena, Jena, Germany). Total SARS-CoV-2 RNA was quantified by quantitative reverse transcription PCR (RT-qPCR), employing the forward primer E_Sarbeco_F1 (ACAGGTACGTTAATAGTTAATAGCGT), the reverse primer E_Sarbeco_R2 (ATATTGCAGCAGTACGCACACA), and the probe E_Sarbeco_P1 (FAM-ACACTAGCCATCCTTACTGCGCTTCG/ZEN/-IBFQ), which targeted the E gene region of SARS-CoV-2[45]. Subgenomic SARS-CoV-2 RNA transcripts were quantified using the same assay, except that the forward primer was substituted with the primer sgLeadSARSCoV2 (CGATCTCTTGTAGATCTGTTCTC), which targeted the leader sequence of the SARS-CoV-2[16]. The assay was performed on a qTower G3 cycler (Analytik Jena) using the NEB Luna Universal Probe One-Step RT-qPCR Kit (New England Biolabs) and the following cycling conditions: 10 min at 55 °C for reverse transcription, 3 min at 94 °C for activation of the polymerase and 40 cycles of 15 s at 94 °C and 30 s at 58 °C.

## Plaque assay and indirect immunofluorescence staining
The number of replication-competent virus particles was determined in 50 mg of lung tissue. For quantification, lung samples were homogenized in a bead mill (Analytik Jena), serially diluted in MEM, and plated on 12-well plates containing confluent Vero E6 cells. After 2.5 h at 37 °C and 5% $CO_2$, the inoculum was removed, and cells were overlaid with Eagle's Minimum Essential Medium (EMEM; Lonza™ BioWhittaker™) medium containing 1.5% microcrystalline cellulose and carboxymethyl cellulose sodium (Vivapur 611p; JRS Pharma). Seventy-two hours after infection the plates were fixed with 4% PBS-buffered formaldehyde.

To conduct indirect immunofluorescence staining, the cells were permeabilized with 0.1% Triton X-100 and blocked for 30 min with 3% BSA diluted in PBS. After washing the plates with PBS, the primary polyclonal anti-SARS Coronavirus nucleocapsid antibody (Invitrogen) was added for 1 h followed by the goat-anti-rabbit IgG-AlexaFluor 488 secondary antibody (Invitrogen) for 45 min. To determine the titers, the plaques were counted using an inverted fluorescence microscope (Axiovert S100, Zeiss).

## Serum-neutralization assay
Neutralizing activity against the SARS-CoV-2 variant B.1 and the Omicron subvariant BA.5 was determined in all hamster sera (0 days post-challenge, 6 dpc. In addition, neutralizing capacity against the Delta variant and the Omicron subvariant BA.1 was tested in day 6 serum samples. Twofold serial dilutions (1:8 to 1:1,024) of complement inactivated (56 °C for 30 min) hamster sera were prepared in 96-well plates. 200 FFU of SARS-CoV-2 diluted in MEM were applied per well and incubated for 1 h at 37 °C. Subsequently, the dilutions were plated on Vero E6 cells cultivated in 96-well plates and incubated for 72 h (B.1, Delta) or 96 h (Omicron BA.1, BA.5) at 37 °C. Thereafter, cells were fixed with PBS-buffered formaldehyde (4%, pH 6.5) and stained with methylene blue (0.75% aqueous solution). Neutralization was considered effective in wells that showed no cytopathic effect. The last neutralized well was reported as the titer. Positive and negative controls were included in all plates. To plot the results, samples without neutralizing activity were set to a titer of 1:4.

### Enzyme-linked immunosorbent assay (ELISA)

SARS-CoV-2-specific IgG levels against the spike protein of the B.1 and BA.5 variants, as well as the nucleocapsid and ORF3a proteins, were measured in hamster sera using an in-house ELISA. Clear 96-well plates with flat bottom (MEDISORP, Thermo Fisher Scientific, catalog number: MW96F) were coated with 5 µl of purified, recombinant, His-tagged SARS-CoV-2 antigens: the spike protein (D614G) of the B.1 variant (Acro Biosystems, catalog number: SPN-C52H3), spike protein of the BA.5.5 variant (GenBank accession: QHD43416, Acro Biosystems, catalog number: SPN-C522p), nucleocapsid protein (GenBank accession: QHD43423, Ray Bio-tech, catalog number: 230-01104), and ORF3a protein (Thermo Fisher Scientific, catalog number: RP-87667). The antigens were diluted in PBS to a final concentration of 20 µg/ml. Additionally, each well was supplemented with 45 µl of coating buffer (50 mM $Na_2CO_3$, 50 mM $NaHCO_3$, pH 9.6). After incubating the plates for 12 h at 4 °C, the plates were washed four times with a washing buffer (0.05% Tween 20 in PBS) and blocked with a blocking buffer (PBS, 1% BSA, 10% FCS) for 1 h. The serum samples were diluted 1:100 in a dilution buffer (PBS, 2% BSA, 0.1% Tween 20) and plated in duplicates of 50 µl per well. The plates were then incubated for 2 h at RT, followed by another washing step. Next, 50 µl of a secondary, horseradish peroxidase (HRP)-conjugated polyclonal goat-anti-hamster IgG (H + L) antibody (Thermo Fisher Scientific, catalog number: PA129626), which was diluted 1:1000 in PBS, was added to each well. After a 1-h incubation at RT, the plates were washed again, and 50 µl of the chromogenic substrate, 3,3′,5,5′-Tetramethylbenzidine (TMB; TCI chemicals, catalog number: T3854) was added to each well. The reaction was stopped with 1 M $H_2SO_4$ after 15 min. The optical density was measured at 450 nm and 570 nm using a SpectraMax Plus 384 plate reader (Molecular Devices).

Moreover, SARS-CoV-2-specific IgA levels against the spike protein of the B.1 and BA.5 variants were measured in nasal washes following the protocol described above with minor adaptions. Nasal washes were diluted 1:50 in a dilution buffer and plated in duplicates. For IgA detection, a polyclonal HRP-conjugated rabbit anti-hamster IgA (Brookwood Biomedical, catalog number: sab3003a) was diluted 1:750 and used as a secondary antibody. After 1 h of incubation at RT, plates were washed, and 50 µl of 1-Step™ Ultra TMB ELISA substrate solution (Thermo Fisher Scientific, catalog number: 34028) was added into each well. The reaction was stopped after 20 min of incubation at room temperature. Positive and negative controls were included in all assays.

### Histopathology and immunohistochemistry

For histopathological analysis, left lung lobes and skinned skulls were fixed in PBS-buffered formaldehyde solution (4%) for 48 h. Skulls were rinsed under tap water for 30 min and decalcified in buffered EDTA solution (pH = 7.0) for three days at 65 °C. Skulls were trimmed to obtain rostral sections at the tip of the first triangular ruga of the hard palate and sections from further caudal at the level of the first molar teeth. Sections of 2 µm thickness were cut from routinely formalin-fixed, paraffin-embedded samples, stained with hematoxylin and eosin, or prepared for immunohistochemistry. Histopathological analyses of lung sections were carried out as described[46]. Nose sections were scored for the presence of lymphocytes, granulocytes, necrosis, epithelial flattening, and loss of cilia as 0 = less than 5% of the epithelium affected, 1 = 5 to 40% of the epithelium affected, 2 = 41 to 80% of the epithelium affected, or 3 = more than 80% of the epithelium affected. Additionally, the airway exudate was characterized. For immunohistochemical analyses, nasal sections were dewaxed in xylene and rehydrated in descending grades of ethanol. Endogenous peroxidase was blocked using $H_2O_2$. Antigen retrieval was achieved by microwaving

sections at 600 W in 750 ml buffered citric acid with 1% Triton X-100 (Roth) for 12 min. The primary monoclonal mouse anti-SARS-CoV/SARS-CoV-2 nucleocapsid protein antibody (Sino Biological, 40143-MM05, dilution: 1:500) was incubated overnight at 4 °C. For universal negative controls, sections were incubated with irrelevant purified mouse IgG (BioGenex) instead of anti-SARS-CoV-2 antibody. Nonspecific binding was blocked with 20% goat serum for 30 min. After washing with PBS/Triton buffer, the secondary antibody, goat-anti-mouse IgG (Vector Laboratories, diluted at 1:200), was applied and incubated for 30 min. The signal was developed with diaminobenzidine tetrahydrochloride (Merck) following 8 min of signal enhancement with Vectastain Elite ABC Kit (Vector Laboratories). Hematoxylin was used as a counterstain. For histopathological evaluation, an Olympus BX41 microscope with a DP80 Microscope Digital Camera (Olympus) and cellSensTM Imaging Software, Version 1.18 (Olympus Soft Imaging Solutions) was used. Automatic digitization was facilitated using an Aperio CS2 slide scanner (Leica Biosystems). Microphotographs were generated with Image Scope Software (Leica Biosystems). Adobe Photoshop or GIMP Software was used to generate figure panels.

### Statistics

Statistical analyses of virological, serological, and histopathological findings were performed with Prism 10.1.0 (GraphPad Software). No data were excluded from the analyses, and no statistical method was used to predetermine sample size. Instead, we selected a sample size based on our previous experience with SARS-CoV-2 vaccination and infection of Syrian hamsters. To adhere to the 3 R principle, we reduced the number of animals used in this study to the minimum that had been experimentally determined in our previous studies[7–10]. We also referred to examples from other studies[47,48]. These publications demonstrate the suitability of the chosen sample size and provide evidence that the number of animals used was the minimum required for the study. A detailed description of statistical analyses performed for each data set can be found in the respective figure legends. Animal trials were conducted in a randomized setup with blinded personnel.

### Reporting summary

Further information on research design is available in the Nature Portfolio Reporting Summary linked to this article.

## Data availability

Datasets generated and/or analyzed during the current study are provided in the main part of the paper or are appended as supplementary data. Source data are provided in this paper.

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

## Acknowledgements

This study was funded by grants from RocketVax Inc., DFG (Deutsche Forschungsgemeinschaft), grant number OS 143/16-1 (to N.O.), and SFB-TR84 Z01b (to A.D.G. and J.T.). We thank the Vaccine Consortium funded by the Swiss National Science Foundation in the framework of the National Research Program "Covid-19" (NRP 78) for productive discussion and exchange of data. We thank C. Thöne-Reineke for their support in animal welfare and husbandry. We thank the European Virus Archive, D. Bourquain from the Robert-Koch-Institut, V. Corman from the Charité Berlin, and C. Reusken from the National Institute for Public Health and the Environment for providing us with SARS-CoV-2 variants used in this study. Vero E6-TMPRSS2 cells were provided by the NIBSC Research Reagent Repository, UK, with thanks to M. Takeda. The European Union's Horizon 2020 Research and Innovation Program—EVA-GLOBAL grant 871029, provided funding for access to SARS-CoV-2 variants. Panels 1a and 5a were created with BioRender.com.

## Author contributions

Du. K. and J.T. conceptualized the project. J.M.A., R.M.V., C.L., D.V., A.A., G.N., Da. K., X.L., A.V., S.K., Du. K. and J.T. conducted the investigations. J.M.A., G.N., H.W., A.V., S.K., Du. K., and J.T. analyzed and visualized the data. A.D.G., H.W., and J.T. provided resources. N.O., A.D.G., and J.T. acquired funding. J.T. administered the project. A.D.G., N.O., Du. K., and J.T. supervised the project. J.M.A., Du. K., and J.T. wrote the original draft. All authors reviewed and edited the manuscript.

## Funding

## Competing interests

Related to this work, Freie Universität Berlin has filed a patent application (PCT/EP2022/051215) for the use of sCPD9 and sCPD9-ΔFCS as vaccines. In this application, J.T., N.O., and D.K. are named as inventors of sCPD9. The patent application does not preclude the use of sCPD9 for scientific purposes. Freie Universität Berlin is collaborating with RocketVax Inc. for further development of sCPD9-ΔFCS as a vaccine and receives funding for research. The remaining authors declare no competing interests.
