## [Peer Review File · Nature Communications]

REVIEWERS' COMMENTS

Reviewer #1 (Remarks to the Author):

I thank the authors for their time and their attention in answering and addressing my concerns. Most concerns have been addressed in the revised manuscript.

Only some minor questions/comments remain:

1. It is unfortunate that the oral swabbing procedure does not allow isolation of live virus. I thank the authors for their explanations on this and hope the authorities may be convinced that oropharyngeal swabbing is allowed in the future to collect better data. Thank you for providing sgRNA data as a surrogate.
2. Regarding the variability in BA.5 shedding. Is sgRNA significantly different for donors? gRNA does not predict actual transmissibility very well. Would the authors see value in adding their commentary into the discussion to highlight the fact that this did not affect transmission. I understand that the scope of this manuscript is not to investigate transmission of BA.5 in detail, but this is nevertheless interesting.

Response to Referees, our answers appear in blue.

Reviewer #1:

I thank the authors for their time and their attention in answering and addressing my concerns. Most concerns have been addressed in the revised manuscript.

We thank the reviewer for their positive evaluation of our revised manuscript and address the remaining comments below.

Only some minor questions/comments remain:

1. It is unfortunate that the oral swabbing procedure does not allow isolation of live virus. I thank the authors for their explanations on this and hope the authorities may be convinced that oropharyngeal swabbing is allowed in the future to collect better data. Thank you for providing sgRNA data as a surrogate.

We agree with the reviewer's assessment. We have observed a drastic increase in regulations pertaining to animal experimentation and a great deal of interference with experimental protocols by state authorities. While strict animal welfare regulations are very important, we do believe that animal experiments are only justified, if as much information as possible can be extracted. This can sometimes be a difficult to resolve situation, we will continue to do our best to convince regulatory authorities to grant reasonable animal use protocols.

2. Regarding the variability in BA.5 shedding. Is sgRNA significantly different for donors? gRNA does not predict actual transmissibility very well. Would the authors see value in adding their commentary into the discussion to highlight the fact that this did not affect transmission. I understand that the scope of this manuscript is not to investigate transmission of BA.5 in detail, but this is nevertheless interesting.

This is an interesting question indeed. We do not observe statistically significant differences in BA.5 sgRNA shedding, however our study was obviously not designed to really address this question. We have now added a short statement in the results section (lines 98-101).